# How Well Prepared Are Dental Students and New Graduates in Pakistan—A Cross-Sectional National Study

**DOI:** 10.3390/ijerph20021506

**Published:** 2023-01-13

**Authors:** Muhammad Qasim Javed, Shazia Nawabi, Usman Anwer Bhatti, Sundus Atique, Mustafa Hussein AlAttas, Ayman M. Abulhamael, Daniel Zahra, Kamran Ali

**Affiliations:** 1Department of Conservative Dental Sciences and Endodontics, College of Dentistry, Qassim University, Buraidah 52571, Saudi Arabia; 2Department of Medical Education, Rawal Institute of Health Sciences, Islamabad 45550, Pakistan; 3Department of Operative Dentistry, College of Dentistry, Riphah International University, Islamabad 45320, Pakistan; 4College of Dental Medicine, Qatar University, QU Health, Doha 2713, Qatar; 5Department of Endodontics, Faculty of Dentistry, King Abdulaziz University, P.O. Box 80209, Jeddah 21589, Saudi Arabia; 6Peninsula Medical School, Faculty of Health (Medicine, Dentistry and Human Sciences), Plymouth University, Plymouth PL48AA, UK

**Keywords:** clinical skills, clinical training, competency, cross-sectional studies, dental education, dentistry, self-perception

## Abstract

The transition of an undergraduate dental student to an actual practicing dentist is a crucial phase and ensuring the preparedness of graduates for the complexity and demands of contemporary dental practice is a challenging task. This study aimed to evaluate the self-perceived preparedness of undergraduate dental students and house officers in the dental colleges of Pakistan. A cross-sectional national study was planned to collect information from dental students and new graduates in Pakistan. The pre-validated Dental Undergraduates Preparedness Assessment Scale (DU-PAS) was used. A purposive sampling technique was utilized to recruit house officers and undergraduate dental students from 27 dental schools in Pakistan. The data analysis was carried out using the R statistical environment for Windows (R Core Team, 2015). A total of 862 responses with 642 females and 219 males were analyzed in the study. Overall, the clinical skills score was 30.56 ± 9.08 and the score for soft skills was 30.54 ± 10.6. The mean age of the participants was 23.42 ± 1.28. Deficiencies were reported in various soft skills and clinical skills attributes. The results highlighted the strengths and weaknesses of dental students and new graduates in Pakistani dental institutions. The findings may be used to further develop and strengthen the teaching and training of dental students in Pakistan.

## 1. Introduction

Dental education is an intricate and arduous process that involves taxing pedagogical experience with unique challenges [1]. The ultimate goal of undergraduate dental education is to equip future dentists with underpinning scientific knowledge, clinical skills and affective skills for the safe practice of Dentistry [2]. In addition to clinical operative procedures, dental graduates are expected to exhibit competence in soft skills that include time management, critical thinking, problem-solving, professionalism, leadership, team working and interprofessional collaborative practice [3]. The aim of undergraduate dental education is to produce scientifically proficient and socio-empathic dental practitioners who strive to observe premier standards of ethics and professionalism to serve society [4,5].

An array of factors influences the quality of undergraduate dental programs including curricular design; clinical training model; effective alignment of program learning outcomes with teaching and assessment methods. A conducive learning environment has a pivotal role in ensuring effective student learning [6]. Likewise, teachers are the backbone of an education system and the provision of quality supervision, support and timely feedback by competent clinical teachers can enhance the students’ experiential learning [7,8]. Previous research on dental students has identified several areas of weaknesses amongst new dental graduates suggesting dental programs may not always be able to train dental students to the expected standards [9,10,11]. The transition from undergraduate dental student to actual dental practice is a crucial but challenging step [12]. Evidence from the literature shows that although most dental students are adequately equipped to carry out simple operative procedures, they may not be prepared to carry out more complex clinical procedures. Considering this, ensuring the preparedness of graduates for the complexity and demands of contemporary dental practice is a daunting task [13]. The dynamic healthcare environment of modern-day society necessitates adequate preparation of the healthcare professions students, hence, enabling them to follow the evidence-based approaches for the prevention, diagnosis and management of oral diseases [4,13].

The dental faculty evaluates students’ work preparedness through continuous formal and informal assessments. Nevertheless, it is important to develop an understanding of work preparedness as perceived by students themselves [6]. Self-reported preparedness may be an important benchmark to identify the shortcomings in training and to rectify them by encompassing a change in curriculum design. Moreover, evaluation of students’ self-perceived preparedness may encourage them to practice reflection through self-assessment that can facilitate the development of lifelong learning skills [8].

Several studies have explored the self-perceived readiness of undergraduate dental students in the United Kingdom, the United States of America, Europe, Hongkong and Malaysia [4,6,12,14,15,16]. Likewise, studies have been conducted in Pakistan [3,17]. However, previous studies in Pakistan were conducted on a limited sample. The number of dental colleges in Pakistan has increased exponentially since 1990 to meet the increasing healthcare needs of the population [18]. Annually, approximately, 3700 dentists graduate from 18 public sector (1200 students) and 43 private sector dental colleges (2500 students) in Pakistan [19].

The aim of the current national study was to evaluate the self-perceived preparedness of undergraduate dental students and new graduates working as house officers in the dental institutions of Pakistan.

## 2. Materials and Methods

A prospective cross-sectional National study was conducted at dental schools in Pakistan. A purposive sampling technique was used to recruit house officers and undergraduate dental students (Pre-final year and Final year) at 27 public and private dental schools in Pakistan by using an e-questionnaire. The total population of the students (Pre-final year and Final year) and house officers in 27 institutes was 5112. The acceptable number of responses was calculated as 588 by setting the confidence level at 99% and the margin of error at 5%. A Raosoft sample size calculator was used (http://www.raosoft.com/samplesize.html, accessed on 23 July 2021). Ethics approval was obtained from the institutional review committee (IRC) at Riphah International University, Pakistan (IIDC/IRC/2021/002/006).

The study employed the Dental Undergraduates Preparedness Assessment Scale (DU-PAS). The DU-PAS is a reliable and validated instrument for measuring a wide range of attributes, competencies and skills expected from dental graduates [20]. The scale has two sections. Section A, the scale comprises 24 items that assess the graduates’ competency in clinical skills. Section B comprises 26 items focusing on soft skills including professionalism, cognitive ability and communication skills. Both sections are scored by using a three-point scale as follows. Items in Section A are related to operative clinical skills which are scored as: (0) No experience, and (1) with a colleague’s practical or/and verbal input, (2) Independently. Items in Section B relate to behavioral and non-operative skills which are scored as: (0) No experience, (1) Mostly, and (2) Always. The cumulative score range for DU-PAS is between 0 and 100.

The e-questionnaire (google form link) and participation information sheet were administered at the culmination of the academic year. The administration was conducted through the focal person at each of the 27 dental schools (seven Public and 20 Private) in the WhatsApp groups of third/final year students and house officers. Informed consent was obtained from all the study participants and the potential participants were informed that the submission of the response will be considered as their consent to participate in the study. Subsequently, four reminders were given at an interval of four weeks. The data collection process was concluded in five months. The data analysis was conducted using the R statistical environment for Windows (R Core Team, 2015). Data collection and analysis were completed in six months. Descriptive statistics were calculated to describe the sample and subgroups, followed by Chi-squared tests of association to compare distributions of responses between subgroups. Where *p*-values are presented, they are derived from Chi-Squared analyses.

## 3. Results

### 3.1. Descriptive Statistics

The total sample consists of 862 responses from 642 (74.59%) Females and 219 (25.41%) Males, with a mean age of 23.42 years (SD = 1.28, Range = 20–30). Whilst mean ages were comparable between genders, the distributions varied significantly between Female (Mean = 23.32, SD = 1.26, Range = 20–30) and Male (Mean = 23.69, SD = 1.31, Range = 20–29) respondents (χ (10, *n* = 855) =19.874, *p* = 0.030)—this is depicted in Figure 1.

Of the 862 responses, 581 (67.40%) were received from Private Institutions and 281 (32.60%) from Public Institutions. In addition, the proportion of Female respondents was significantly higher at Private (77.28%) than Public (69.04%) institutions (*p* = 0.012). However, the distribution of Age within Private and Public Institutions did not differ significantly (*p* = 0.129). The sample included 507 (58.83%) final-year BDS students (359 Female, 148 Male), 36 (4.18%) pre-final-year BDS students (25 Female, 11 Male), and 319 (37.01%) House Officers (259 Female, 60 Male). The majority of the respondents were from Punjab (*n* = 483, 56.03%), followed by Federal Capital Territory (*n* = 168, 19.49%), Khyber Pakhtunkhwa (*n* = 142, 16.47%), Sindh (*n* = 53, 6.15%), and Baluchistan (*n* = 16, 1.86%).

### 3.2. Overall Scores

Overall scores by Part are shown in Table 1. These are calculated based on converting responses in Part A (24 Items) and B (26 Items) to numeric scores as; Independently = 2, With Help = 1, and No Experience = 0 (Part A) and Always = 2, Mostly = 1, and No Experience = 0 (Part B).

#### 3.2.1. Part A Items

##### By Gender

The responses to each item, by gender, are shown in Figure 2. This figure is ordered from the highest to lowest proportion of ‘independent’ responses and highlights items for which Chi-squared analyses suggest the profile differs significantly by gender at the *p* < 0.05 level. These items are also listed in Table 2 along with related *p*-values.

##### By Institution Category

The responses to each item, by Institution Type, are shown in Figure 3. This figure is ordered from the highest to lowest proportion of responses reported as “independently,” and highlights items for which the response profile differs significantly between Private and Public Institutions (*p* < 0.05). These items are also listed in Table 3 along with the related *p*-values.

#### 3.2.2. Part B Items

##### By Gender

The responses to each item, by gender, are shown in Figure 4. This figure is ordered from the highest to lowest proportion of ‘Always’ responses and highlights items for which the response profile differs significantly by gender at the *p* < 0.05 level. These items are also listed in Table 4, along with related *p*-values.

##### By Institution Category

The responses to each item, by Institution Category, are shown in Figure 5. This figure is ordered from highest to lowest proportion of ‘Always’ responses and highlights items for which the response profile differs significantly between Private and Public/Government Institutions at the *p* < 0.05 level. These items are also listed in Table 5, along with related *p*-values.

## 4. Discussion

This is the first national study reporting the self-perceived preparedness of dental students and new graduates from multiple private and public sector dental institutions in Pakistan. The study provides useful insights into the preparedness of dental students and graduates from Pakistan and highlights several areas of weakness that warrant attention and remedial measures by dental educators in Pakistan.

Based on the total mean score of the study respondents, dental undergraduates in Pakistan reported being less prepared (mean score 61.10%) compared to dental students from other countries such as Malaysia (79.5%) and the United Kingdom (74%) [4,14]. These findings may be attributed to gaps in the quality assurance of undergraduate dental education in Pakistan. The number of dental institutions has grown exponentially in Pakistan over the last decade. However, the resources of the Pakistan Medical Commission (PMC) which regulates dental education in Pakistan have not increased commensurately and this may have resulted in deficiencies in the close monitoring of undergraduate dental programs. Moreover, the data collection for this study was carried out during the COVID-19 pandemic which may have also influenced the teaching and training of dental students adversely

In the present study, respondents felt most prepared to record medical history, remove dental caries and administer inferior dental nerve blocks. Similar findings were reported by other studies investigating students from not only Pakistan but also the United Kingdom [3,14,17]. Considering these are generally straightforward, it is not surprising to find this similarity among students of developing and developed countries.

Regarding areas of deficiency in clinical skills, experience in performing endodontics was observed to be deficient and respondents from both private as well as public sector institutions reported a lack of experience in performing endodontic treatment on multi-rooted teeth (61.1%) and also single-rooted teeth (33.3%). In contrast, Ali et al., reported only that only 2.4% of dental students in the UK did not have any experience in carrying out endodontic treatment on single-rooted teeth [14]. Moreover, with respect to the endodontic treatment of multi-rooted teeth, a significant difference was observed in the preparedness of respondents from the private sector compared with those from the public sector. The differences may be attributed to a deficiency of specialist endodontists and the limited availability of suitable patients requiring endodontics in private institutions. These factors might have impacted the teaching of undergraduate students adversely. These differences are quite remarkable and highlight the need for further clinical experience in endodontics for dental students in Pakistan [21]. Interestingly, developed countries face a similar challenge in undergraduate endodontic education [22,23].

Radiography skills were also flagged up as a relatively weak area in the present study. Respondents of the present study were the least prepared to take bitewing radiographs in contrast to participants from the UK and Malaysia [4,14]. Qazi et al. reported a similar lack of confidence in performing bitewings among Pakistani students in contrast to students from Malaysia and the United Kingdom [4,14,17]. Anecdotal evidence suggests that dental students in Pakistan do not obtain adequate experience in bitewing radiography. Instead, there is a preference for using periapical radiographs to diagnose dental caries which is inappropriate [3]. These findings underscore the need to revisit radiology teaching and align it with contemporary and evidence-based clinical practice.

Other areas of reported weakness include assessment of orthodontic treatment needs; provision of crowns and, the fabrication of cast partial dentures. These findings concord with those reported by Qazi et al., Ali et al. and Mat Yudin et al., who reported similar challenges for students in Pakistan, the United Kingdom and Malaysia, respectively [4,14,17]. More importantly, however, there was a significantly high proportion of participants in this study who reported “no experience” in these skills. Given these are core skills expected from a general dental practitioner, these findings raise serious concerns regarding the breadth of clinical training in Pakistani dental institutions. The present study has managed to capture the magnitude of these deficiencies in clinical skills which was not achieved in a previous study on a relatively smaller study sample.

The majority of the respondents felt prepared to communicate with the patients, seek help from supervisors, and protect patient confidentiality among other attributes of professionalism. These observations are positive and demonstrate a professional and ethical culture in educational and clinical settings. However, the participants felt less prepared regarding the interpretation of research to inform their clinical practice, the evaluation of dental materials/products using an evidence-based approach, and the referral of patients suspected of oral cancer. Similar observations have been reported in previous studies on the preparedness of dental students and new graduates [3,14,17]. Understanding and applying evidence-based dentistry in undergraduate dental education remains a challenge globally despite the growing emphasis on its importance in the last two decades [24]. In the absence of a structured course on evidence-based dentistry in undergraduate dental curricula in Pakistan, the findings are not unexpected. Dental institutions in Pakistan need to develop a comprehensive strategy to incorporate principles of evidence-based dentistry in undergraduate curricula [25,26]. It may be helpful to learn from the experiences of dental institutions in the West that provide teaching on evidence-based practice to make informed decisions on updating existing dental curricula in Pakistan [27,28].

Given that Pakistan has one of the highest global incidences of oral cancer, it is ironic that Pakistani dental students and graduates feel unprepared to recognize and refer suspected oral cancer. This may be related to a lack of consistency in the teaching and clinical exposure to patients with oral cancer. Early detection and prompt management of oral cancer are most critical in improving cancer survival rates and reducing cancer-related morbidity and mortality [29]. Unfortunately, a similar lack of preparedness was identified in other countries such as the United Kingdom and Malaysia [4,14]. Structured exposure to oral cancer patients in specialist oral and maxillofacial surgery settings is essential to improve students’ confidence in the recognition and referral of oral cancer.

The data also highlighted several gender-related differences in self-reported preparedness. Female students felt better prepared in history taking, clinical examination, radiographic interpretation, treatment sequencing and non-surgical periodontal treatment. On the other hand, male respondents reported being better prepared to assess orthodontic treatment needs and bitewing radiography. Other notable differences related to affective skills as female respondents felt more prepared to maintain professional relationships with patients, effectively communicate with patients and reflect on their clinical practice. A similar gender-related disparity is observed in other studies, with female students reported to demonstrate better empathy towards patients compared to male students [30]. In contrast, male respondents felt more prepared to interpret research for clinical application and evaluate materials using an evidence-based approach. However, contrary to the evidence indicating a lack of confidence among female undergraduate dental students, the findings of our study show female students as more prepared than male students in terms of clinical skills and professionalism [12,30]. These differences may be attributed to a higher proportion of females in the study sample and perhaps also to growing female empowerment in society [31,32].

Finally, differences were also noted between participants from private and public sector institutions. Participants from private institutions felt better prepared in attributes related to professionalism which may reflect cultural differences in private and public sector institutions. Additionally, Public sector institutions face a much higher inflow of patients. Increased workload and time constraints may adversely impact novice clinicians’ interpersonal skills, although this cannot be justified and needs remedial action.

Preparedness for practice is a complex concept, which can be studied from the perspective of academicians, students, dentists, and patients [33,34]. In this study, the preparedness of a large sample was measured from a student’s perspective using a validated tool which contributes to improved internal and external validity. The main limitation of this study is that the results are based on the self-evaluation of the students. Therefore, caution should be exercised while interpreting self-reported preparedness by dental students as the scores may be inflated by “unconscious incompetence” or a lack of experience in self-evaluation [35,36]. It is acknowledged that the evidence regarding the correlation between actual clinical competence and perceived self-confidence is weak [37]. Nevertheless, the overall mean scores of participants in this study appear to be lower than those reported in previous studies from the UK, Malaysia, and Pakistan which also used the DU-PAS instrument. These differences may underscore “conscious incompetence” amongst the participants in this study or a genuine lack of preparedness. Another factor that may have contributed to the lower mean scores of the study participants might be related to the inclusion of Pre-final year students who had less experience. It is also worth emphasizing that the study was conducted following the start of the COVID-19 pandemic, and it is possible that the lower level of preparedness reported by the study participants may reflect the adverse impact of the pandemic on their teaching and training [38,39,40]. Dental institutions must ensure that appropriate checks are in place to monitor the quality of their graduates and that support is available for new graduates to address any gaps in clinical experience. The implementation of reflective learning with feedback in the undergraduate Pakistani curriculum can be a step towards improving the self-perceived preparedness of clinical skills [41].

## 5. Conclusions

This study investigated the self-reported preparedness of a large sample of Pakistani undergraduate dental students and new graduates using a validated instrument. The results highlight the strengths and weaknesses of dental students and new graduates in Pakistani dental institutions. The findings may be used to further develop and strengthen the teaching and training of dental students in Pakistan. The findings may be of interest to dental educators in Pakistan and further afield. DU-PAS appears to be a reliable instrument to quantify the preparedness of dental graduates from different parts of the world. In the future, mixed methods research involving qualitative methods may be employed to engage with a range of stakeholders in dental education. Such an approach may help gain a deeper understanding of the factors which influence the preparedness of dental students, and the role dental institutions can play to improve the learning experiences of the students.

## Figures and Tables

**Figure 1 ijerph-20-01506-f001:**
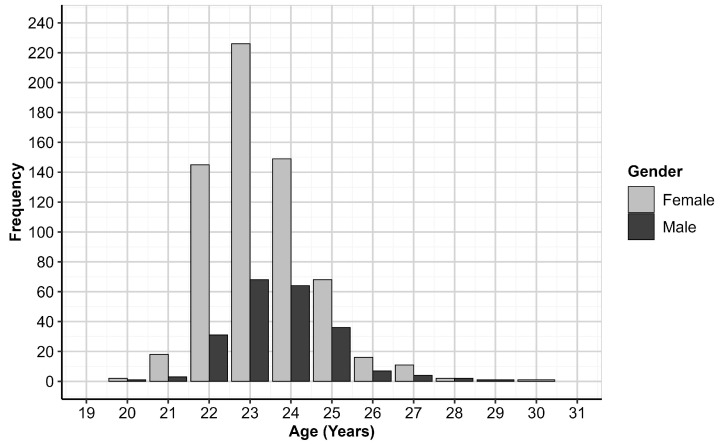
Distribution of Age by Gender.

**Figure 2 ijerph-20-01506-f002:**
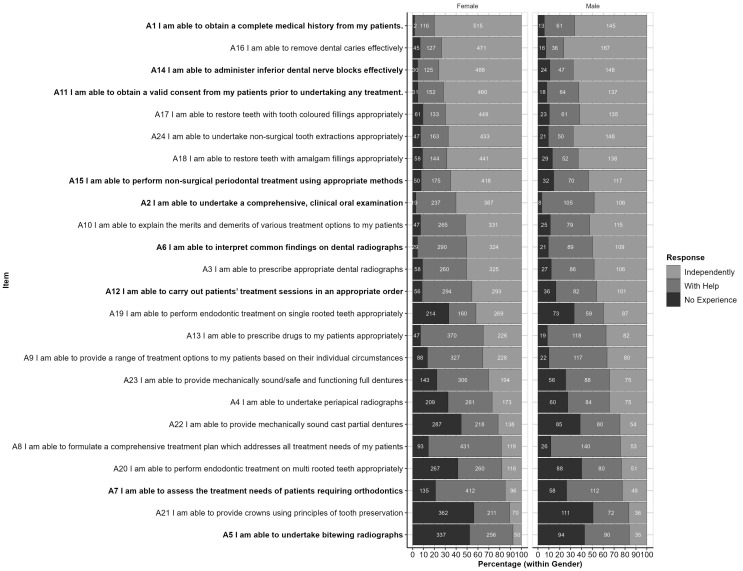
Responses by Part A items (within Gender): Reponses are ordered from highest to lowest proportion of ‘Independent’ responses (across Genders). Items for which response profiles differ significantly by gender (*p* < 0.05) are highlighted in bold text.

**Figure 3 ijerph-20-01506-f003:**
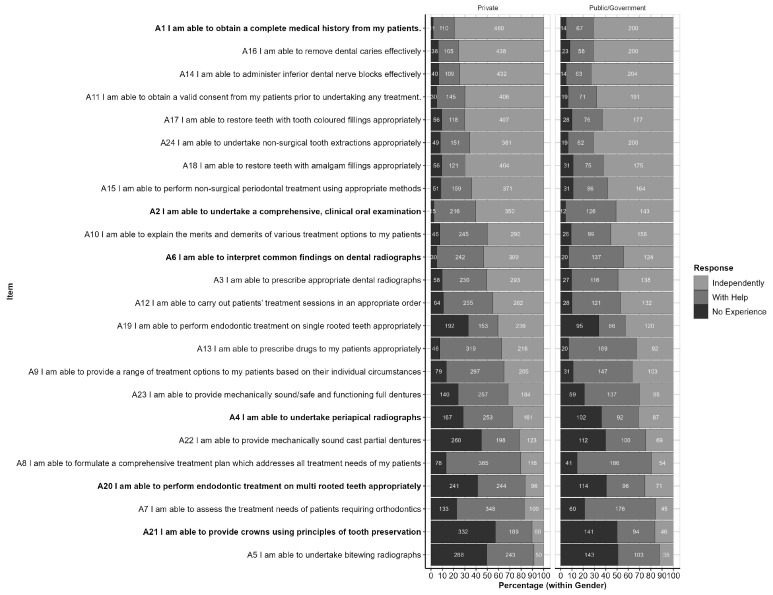
Responses by Part A items (within Institution Category); Responses are ordered from highest to lowest proportion of ‘Independent’ responses (across Institution Categories). Items for which response profiles differ by Institution Category (*p* < 0.05) are highlighted in bold text.

**Figure 4 ijerph-20-01506-f004:**
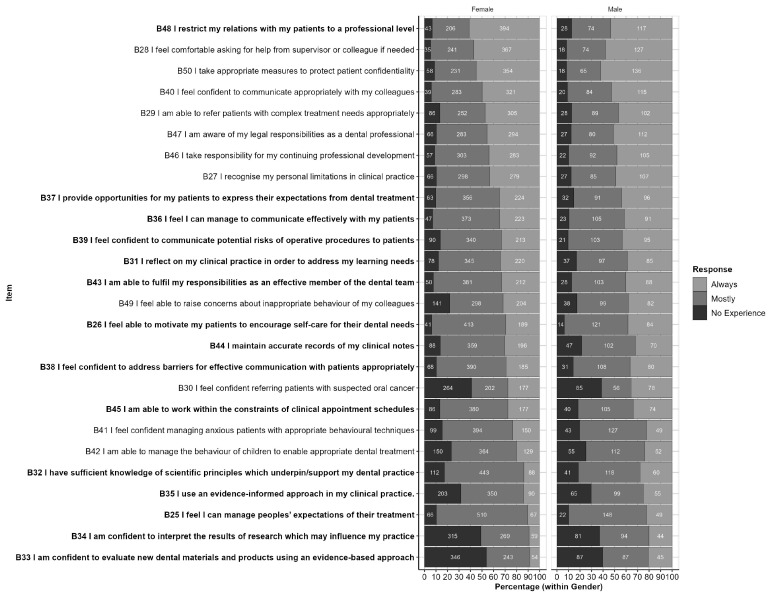
Responses by Part B items (within Gender): Responses are ordered from highest to lowest proportion of ‘Always’ responses (across Genders). Items for which response profiles differ by gender (*p* < 0.05) are highlighted in bold text.

**Figure 5 ijerph-20-01506-f005:**
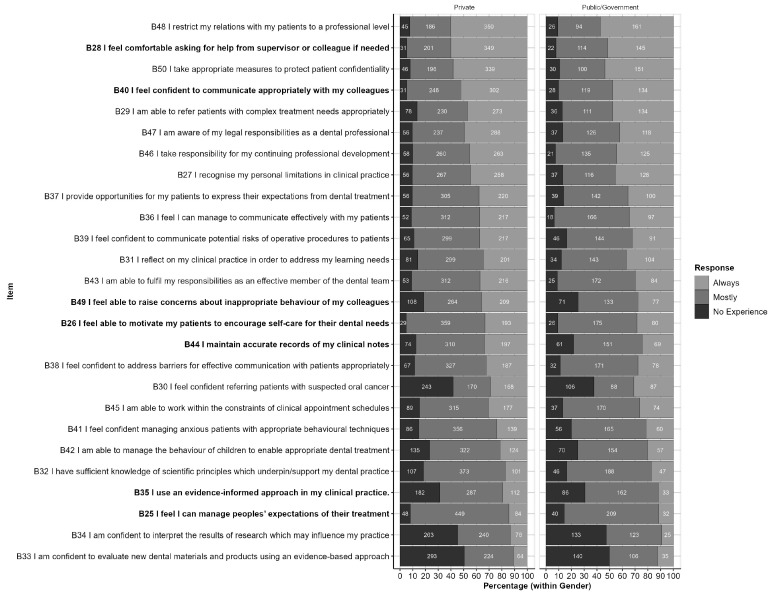
Responses by Part B items (within Institution Category): Responses are ordered from highest to lowest proportion of ‘Always’ responses (across Institution Categories). Items for which response profiles differ by Institution Category at the *p* < 0.05 level are high.

**Table 1 ijerph-20-01506-t001:** Mean Total Score of Participants.

Part	*n*	Mean	SD	Min.	Max.	Range	IQR
A (24 Items)	862	30.56	9.08	0	48	48	12
B (26 Items)	862	30.54	10.6	0	52	52	13

**Table 2 ijerph-20-01506-t002:** Association between Gender and Item Responses for Part A: Specific *p*-values from Chi-Squared tests of association for items showing a statistically significant association between Gender and Item Responses for Part A.

Item	*p*-Value
A1 I am able to obtain a complete medical history from my patients	<0.001
A14 I am able to administer inferior dental nerve blocks effectively	0.002
A11 I am able to obtain a valid consent from my patients prior to undertaking any treatment	0.027
A15 I am able to perform non-surgical periodontal treatment using appropriate methods	0.002
A2 I am able to undertake a comprehensive, clinical oral examination	0.010
A6 I am able to interpret common findings on dental radiographs	0.018
A12 I am able to carry out patients’ treatment sessions in an appropriate order	0.003
A7 I am able to assess the treatment needs of patients requiring orthodontics	0.002
A5 I am able to undertake bitewing radiographs	0.001

**Table 3 ijerph-20-01506-t003:** Association between Institution Type and Item Responses for Part A. Specific *p*-values from Chi-Squared tests of association for items showing a statistically significant association between Institution Type and Item Responses for Part A.

Item	*p*-Value
A1 I am able to obtain a complete medical history from my patients	0.007
A2 I am able to undertake a comprehensive, clinical oral examination	0.024
A6 I am able to interpret common findings on dental radiographs	0.039
A4 I am able to undertake periapical radiographs	0.008
A20 I am able to perform endodontic treatment on multi rooted teeth appropriately	0.005
A21 I am able to provide crowns using principles of tooth preservation	0.025

**Table 4 ijerph-20-01506-t004:** Association between Gender and Item Responses for Part B: Specific *p*-values from Chi-Squared tests of association for items showing a statistically significant association between Gender and Item Responses for Part B.

Item	*p*-Value
B48 I restrict my relations with my patients to a professional level	0.010
B37 I provide opportunities for my patients to express their expectations from dental treatment	0.001
B36 I feel I can manage to communicate effectively with my patients	0.028
B39 I feel confident to communicate potential risks of operative procedures to patients	0.015
B31 I reflect on my clinical practice in order to address my learning needs	0.038
B43 I am able to fulfil my responsibilities as an effective member of the dental team	0.003
B26 I feel able to motivate my patients to encourage self-care for their dental needs	0.044
B44 I maintain accurate records of my clinical notes	0.011
B38 I feel confident to address barriers for effective communication with patients appropriately	0.013
B45 I am able to work within the constraints of clinical appointment schedules	0.014
B32 I have sufficient knowledge of scientific principles which underpin/support my dental practice	0.000
B35 I use an evidence-informed approach in my clinical practice	0.001
B25 I feel I can manage peoples’ expectations of their treatment	<0.001
B34 I am confident to interpret the results of research which may influence my practice	<0.001
B33 I am confident to evaluate new dental materials and products using an evidence-based approach	<0.001

**Table 5 ijerph-20-01506-t005:** Association between Institution Category and Item Responses for Part B Specific *p*-values from Chi-Squared tests of association for items showing a statistically significant association between Institution Category and Item Responses for Part B.

Item	*p*-Value
B28 I feel comfortable asking for help from supervisor or colleague if needed	0.047
B40 I feel confident to communicate appropriately with my colleagues	0.036
B49 I feel able to raise concerns about inappropriate behaviour of my colleagues	0.015
B26 I feel able to motivate my patients to encourage self-care for their dental needs	0.034
B44 I maintain accurate records of my clinical notes	0.001
B35 I use an evidence-informed approach in my clinical practice	0.012
B25 I feel I can manage peoples’ expectations of their treatment	0.017

## Data Availability

Data will be made available upon request.

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
