# Peer review of "How Well Prepared Are Dental Students and New Graduates in Pakistan—A Cross-Sectional National Study"

_ijerph, 2023, doi:10.3390/ijerph20021506_

Round 1

Reviewer 1 Report

Abstract

The abstract is concise and gives a clear idea of the overall study.

Introduction

The introduction is well-organized and well-written with a good flow. Since similar studies have already been conducted before in Pakistan, why would the authors choose to do it again? If proper sample size calculations were done in those studies, the results would be valid and generalisable.

Material and Methods

-The sample size calculation is not wrong. But since past studies are available, the authors could have used the previous studies’ preparedness level of dental students in Pakistan. This may result in a smaller sample size needed.

-Please add a line in the data analysis paragraph on the different tests that will be used to interpret the data.

Discussion

Line 221: It would be good if the authors can explain or even speculate why dental undergraduates of Pakistan were less prepared compared to dental students from other countries.

Line 233. Once again, it would be good if the authors can explain why was there a significant difference in preparedness level between participants from private and government institutions. (Number of patients attending the clinic? Charges??)

Line 245 please provide reference to support your statement.

Please add a paragraph on the limitation of the study.

Author Response

Thanks a lot for reviewing out manuscript. Please find attached word file for responses

Reviewer 2 Report

The study is interesting, but I suggest reviewing some aspects:

1. Figures 2-3-4-5 are important, but I think they are not of sufficient quality to be published. I suggest looking for another type of edition that improves the quality.

2. In the Discussion (line 220), it is said that Pakistan's mean score is 61.10 and Malaysia's is 79.5%. I suggest clarifying whether all scores will be % or not.

3. In the explanation of the Method it is difficult to understand which test is used to compare gender or institutions, although it is specified later in the legend of Table 2 and 3 (lines 159 and 175). It should be clarified in the text (line 153).

4. When trying to reproduce the calculation of the p value by means of a calculator, it is observed that question A2 of the analysis by Institution Category is 0.00024, and not 0.024 (line 179).

5. The formula χ(1n=862) and χ(10, n=855) of lines 133-135 is not well understood. I suggest clarifying.

6. In the Method it is said that “Both sections are scored by using a three-point scale as follow. Section A: (0) No experience, and (1) with colleague's practical or/and verbal input, (2) Independently”. These answers are not well understood. I suggest clarifying the scoring for those who are not familiar with the DU-PAS test (line 103-107).

Author Response

(The authors gave the same response as above.)
